# Processing of different spatial scales in the human brain

**Michael Peer[1,2,3]\*, Yorai Ron[1,2], Rotem Monsa[1,2], Shahar Arzy[1,2]**

[1]Department of Medical Neurosciences, Faculty of Medicine, Hebrew University of Jerusalem, Jerusalem, Israel; [2]Department of Neurology, Hadassah Hebrew University Medical School, Jerusalem, Israel; [3]Department of Psychology, University of Pennsylvania, Philadelphia, United States

**Abstract** Humans navigate across a range of spatial scales, from rooms to continents, but the brain systems underlying spatial cognition are usually investigated only in small-scale environments. Do the same brain systems represent and process larger spaces? Here we asked subjects to compare distances between real-world items at six different spatial scales (room, building, neighborhood, city, country, continent) under functional MRI. Cortical activity showed a gradual progression from small to large scale processing, along three gradients extending anteriorly from the parahippocampal place area (PPA), retrosplenial complex (RSC) and occipital place area (OPA), and along the hippocampus posterior-anterior axis. Each of the cortical gradients overlapped with the visual system posteriorly and the default-mode network (DMN) anteriorly. These results suggest a progression from concrete to abstract processing with increasing spatial scale, and offer a new organizational framework for the brain's spatial system, that may also apply to conceptual spaces beyond the spatial domain.
DOI: https://doi.org/10.7554/eLife.47492.001

## Introduction

Over the past few decades, research of the brain's spatial system advanced tremendously, providing insights into how the brain represents complex information and how these processes are impaired in disease states (e.g. *Banino et al., 2018*; *Kunz et al., 2015*; for reviews see *Buzsáki and Moser, 2013*; *Epstein et al., 2017*; *Moser et al., 2008*). However, scientific investigations of spatial cognition in humans and animals are often limited to small scale environments such as single rooms or short walkable pathways. It is therefore unclear whether representation and processing of large-scale environments rely on the same neurocognitive systems (*Wolbers and Wiener, 2014*). This question is of importance for several reasons. First, the lack of knowledge on how the brain's spatial system treats different spatial scales affects interpretation of past investigations that used different types of experimental environments. Second, disorientation is a prevalent symptom across neurological and psychiatric disorders, but remains poorly understood and diagnosed, in part because it may have different subtypes that manifest at different spatial scales (*Peer et al., 2014*). Finally, recent findings suggest that the brain's spatial system is also used to represent conceptual knowledge (*Behrens et al., 2018*; *Bellmund et al., 2018*; *Constantinescu et al., 2016*; *Gärdenfors, 2000*). Since large-scale environments are often remembered in a schematic manner not consistent with Euclidean geometry (*McNamara, 1986*; *Moar and Bower, 1983*; *Tversky, 1981*), understanding their representation may provide clues to representation of abstract domains.

Previous neuroscientific evidence supports the idea that the brain's spatial representations are not unified but separated into multiple scales. Functional MRI studies in humans demonstrated that locations within rooms and their surrounding buildings are coded in different cortical regions (*Kim and Maguire, 2018*), and that directions are represented in the retrosplenial complex with

**\*For correspondence:**
michael.peer@mail.huji.ac.il

**Competing interests:** The authors declare that no competing interests exist.

respect to the local axis of a room irrespective of its large-scale context (*Marchette et al., 2014*). Electrophysiological evidence in animals also points to separate representation of small scale regions and their large-scale context, as grid- and place-cells within the medial temporal lobe undergo remapping when crossing borders between rooms (*Fyhn et al., 2007*; *Skaggs and McNaughton, 1998*; *Tanila, 1999*), and form independent representations of different segments of the environment (*Derdikman et al., 2009*; *Derdikman and Moser, 2010*; *Paz-Villagrán et al., 2004*; *Spiers et al., 2015*). Recordings from the rat retrosplenial cortex also demonstrate coding of location both in the immediate small-scale region and in the large-scale surrounding environment (*Alexander and Nitz, 2017*; *Alexander and Nitz, 2015*). Finally, evidence from patients with disorientation disorders shows that disorientation can be limited to a specific spatial scale according to the underlying lesion (*Peer et al., 2014*). Patients with lateral parietal cortex lesions are impaired in navigating their immediate, small-scale environment ('egocentric disorientation'; *Aguirre and D'Esposito, 1999*; *Stark, 1996*; *Wilson et al., 2005*). In contrast, patients with retrosplenial lesions (*Aguirre and D'Esposito, 1999*; *Takahashi et al., 1997*) and Alzheimer's disease (*Monacelli et al., 2003*; *Peters-Founshtein et al., 2018*) show the opposite pattern – correct localization in the immediately visible environment but inability to navigate in the larger unseen environment. Despite this evidence, few neuroscientific studies directly contrasted between representation of different scales of space. Several studies indicated a posterior-to-anterior progression from small to large scales along the hippocampal axis, manifested as larger spatial receptive fields, in both humans and animals (*Brunec et al., 2018*; *Kjelstrup et al., 2008*; *Poppenk et al., 2013*). However, these investigations only used routes ranging up to several meters, and focused only on the hippocampus and not on the rest of the brain's spatial system. Another fMRI study contrasted coarse- and fine-grained spatial judgments in one scale (city), finding increased hippocampal activity for fine-grained distinctions (*Hirshhorn et al., 2012a*). In the current work, we sought to characterize human brain activity under ecological experimental settings, across a large range of spatial scales, when directly manipulating only the parameter of spatial scale. To this aim, we asked subjects to compare distances between real-world, personally familiar locations across six spatial scales (rooms, buildings, neighborhoods, cities, countries and continents; *Figure 1*), under functional MRI, and looked for differences in brain response for the different scales.

## Results

### Posterior-anterior gradients of spatial scale selectivity

To investigate spatial scale-selective activity, we looked for voxels showing difference in response to task performance at the different scales, and characterized their gradual response profiles by fitting a Gaussian function to the beta value graphs at each voxel (*Figure 2—figure supplement 1*). This analysis identified three cortical regions that displayed a continuous gradual shift in spatial scale selectivity: the medial temporal cortex, medial parietal cortex and lateral parieto-occipital cortex (*Figure 2A–D*, *Figure 2—figure supplement 2*). Activity in these regions displayed a gradual shift from selectivity for the smallest spatial scales (room, building) in their posterior parts, followed by selectivity for medium scales (neighborhood, city) more anteriorly, and for the largest scales (country, continent) in the most anterior part of each gradient (*Figure 2E*; $p < 0.001$ for all gradients, permutation test on linear fit slope, FDR-corrected). The three scale-selective gradients were symmetric across the two hemispheres. Extraction of the scale with maximal response from each voxel (while disregarding the pattern of activity at other scales) also demonstrated posterior-to-anterior progression along the three abovementioned gradients (*Figure 2E*, *Figure 2—figure supplement 3*; $p < 0.001$ for all gradients, permutation test on linear fit slope, FDR-corrected). To further characterize the scale selectivity of each region, we plotted the event-related activity and beta values for each spatial scale at each part of the three gradients. Results showed the same gradual posterior-anterior shift from small to large spatial scales, with each part of the gradient having a preferred scale and gradually diminishing activity to other scales around it (*Figure 2—figure supplement 4A–C*). Finally, in light of previous findings of spatial scale selectivity changes along the hippocampal long axis (*Brunec et al., 2018*; *Poppenk et al., 2013*), we measured average spatial scale selectivity along the hippocampus. Activity shifted from small to large scales along the posterior-anterior axis of the hippocampus (*Figure 2E*; $p < 0.001$ for average position of Gaussian fit peak, permutation test on linear

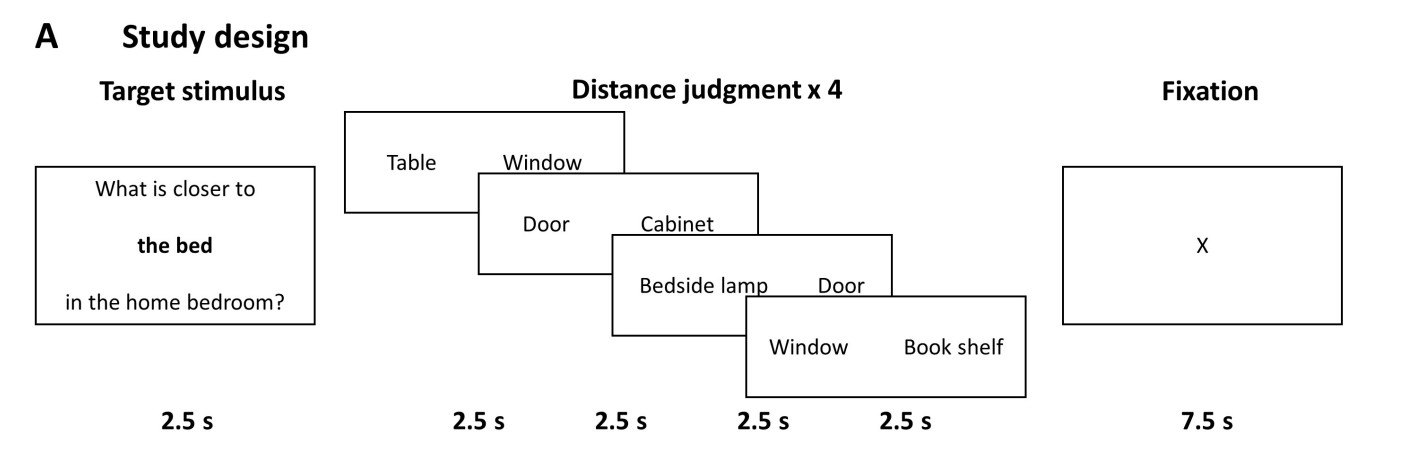

**Figure 1.** Study design and stimuli. (**A**) The design of the study. In each block, subjects viewed one target item in a specific scale and location, and then performed four proximity comparisons for pairs of other items from the same location. All stimuli were provided by the subjects from locations personally familiar to them, and target and comparison items were chosen randomly from the subject's stimulus set. (**B**) Examples of stimuli (subject-provided locations and items) in each spatial scale.

DOI: https://doi.org/10.7554/eLife.47492.002

The following figure supplements are available for figure 1:

**Figure supplement 1.** increase in size of spatial scales.

DOI: https://doi.org/10.7554/eLife.47492.003

**Figure supplement 2.** Behavioral results.

DOI: https://doi.org/10.7554/eLife.47492.004

fit slope, FDR-corrected). Using the same analysis at the individual subject level, 16 of 19 subjects showed significant increase in preferred scale along the lateral parietal gradient, 17 of 19 along the medial temporal gradient, 17 of 19 along the medial parietal gradient, and 6 of 19 along the hippocampus (all $p < 0.05$, permutation test on linear fit slope, FDR-corrected).

In addition to the continuous gradients, several other brain regions displayed scale-specific activity not organized as a continuous gradient (*Figure 3*, *Supplementary file 1*). Clusters of activity at the supramarginal gyrus, posterior temporal cortex, superior frontal gyrus and dorsal precuneus displayed the highest activity levels for the smallest spatial scales (room and building), and their activity gradually diminished for larger scales (*Figure 2—figure supplement 4D*). In contrast, the lateral occipital cortex and the anterior medial prefrontal cortex clusters displayed the opposite pattern of

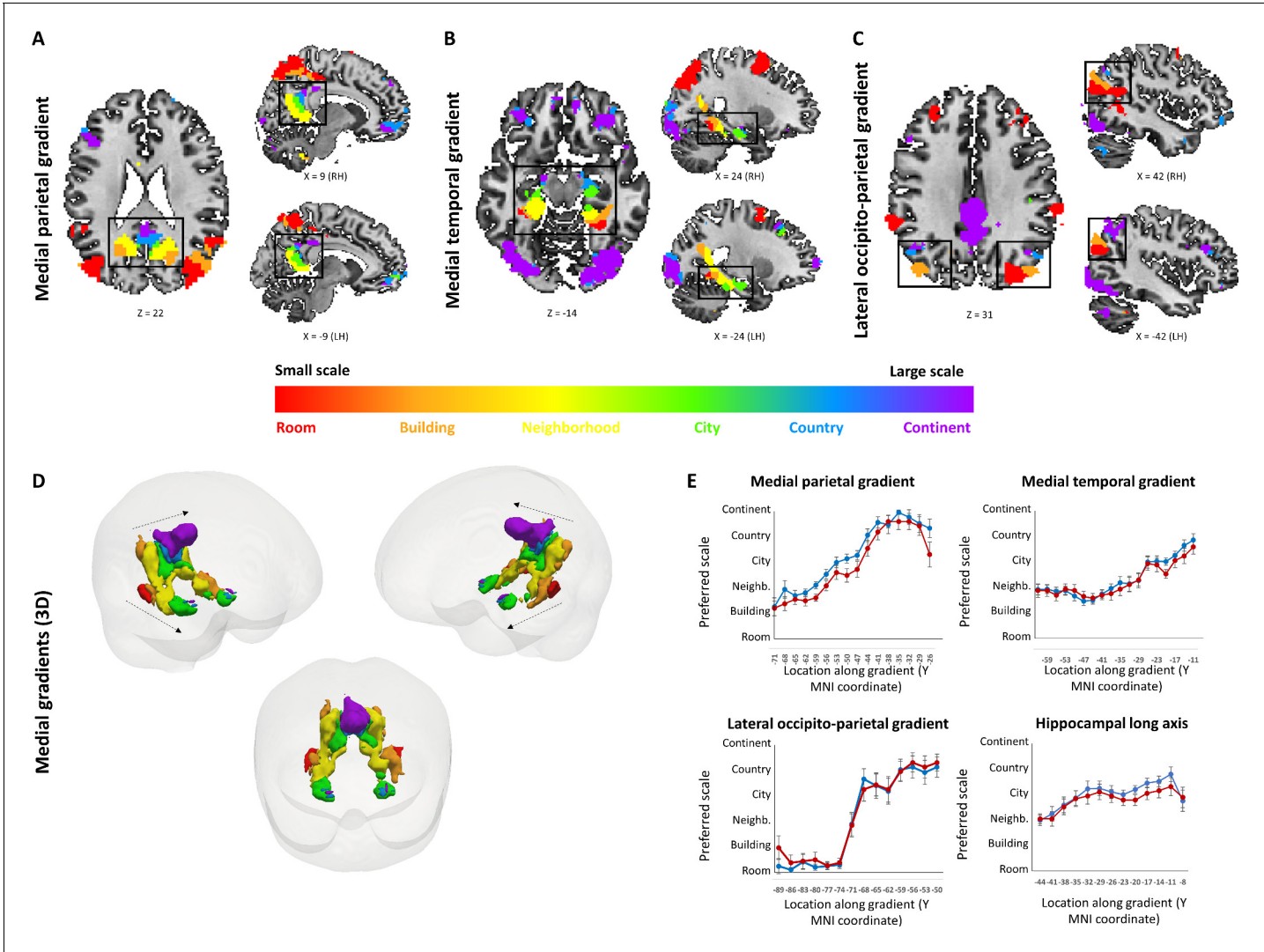

**Figure 2.** Small to large spatial scales preferentially activate regions along continuous posterior-anterior gradients. Three cortical gradients were observed demonstrating a continuous shift in spatial scale selectivity. Within each gradient, posterior regions were selectively active for smaller spatial scales, and anterior ones for larger spatial scales. Colors indicate Gaussian fit peak scale position (voxels identified by ANOVA across beta values, p<0.01, FDR-corrected for multiple comparisons, minimum r2 of fit = 0.7). (A) Medial parietal gradient, (B) Medial temporal gradient, (C) lateral occipito-parietal gradient. (D) 3D visualization of the two medial gradients (gradients marked by dashed arrows, other activations not shown). (E) change in average spatial scale selectivity along the posterior-anterior axis of each gradient and along the hippocampal long axis (X axis represents MNI coordinates from posterior to anterior, blue – average position of a Gaussian fit peak for all scale-sensitive voxels at each coordinate, red – average position of scale with maximum activity for all scale-sensitive voxels at each coordinate). RH – right hemisphere, LH – left hemisphere. Full volume maps of these results are available online at https://github.com/CompuNeuroPsychiatryLabEinKerem/publications_data/tree/master/spatial_scales (*Peer et al., 2019*, copy archived at https://github.com/elifesciences-publications/publications_data).
DOI: https://doi.org/10.7554/eLife.47492.005

The following figure supplements are available for figure 2:

**Figure supplement 1.** Main data analysis pipeline.
DOI: https://doi.org/10.7554/eLife.47492.006
**Figure supplement 2.** Volume view of all scene-selective activations.
DOI: https://doi.org/10.7554/eLife.47492.007
**Figure supplement 3.** Gradients of spatial scale selectivity – scale with highest activity (beta value) at each voxel.
DOI: https://doi.org/10.7554/eLife.47492.008
**Figure supplement 4.** Activity profiles for spatial scale-sensitive regions.
DOI: https://doi.org/10.7554/eLife.47492.009
**Figure supplement 5.** Effects of different potential contributing factors.
*Figure 2 continued on next page*

*Figure 2 continued*

DOI: https://doi.org/10.7554/eLife.47492.010

higher activity for the largest spatial scales (city, country and continent), and gradually decreasing activity for the smaller scales (*Figure 2—figure supplement 4D*).

### The three cortical scale-selective gradients extend anteriorly from scene-responsive cortical regions

The three cortical gradients identified by our analyses are located in close proximity to known scene-responsive cortical regions – parahippocampal place area (PPA), retrosplenial complex (RSC) and occipital place area (OPA) (*Epstein et al., 2017*). To test the exact locations of these regions

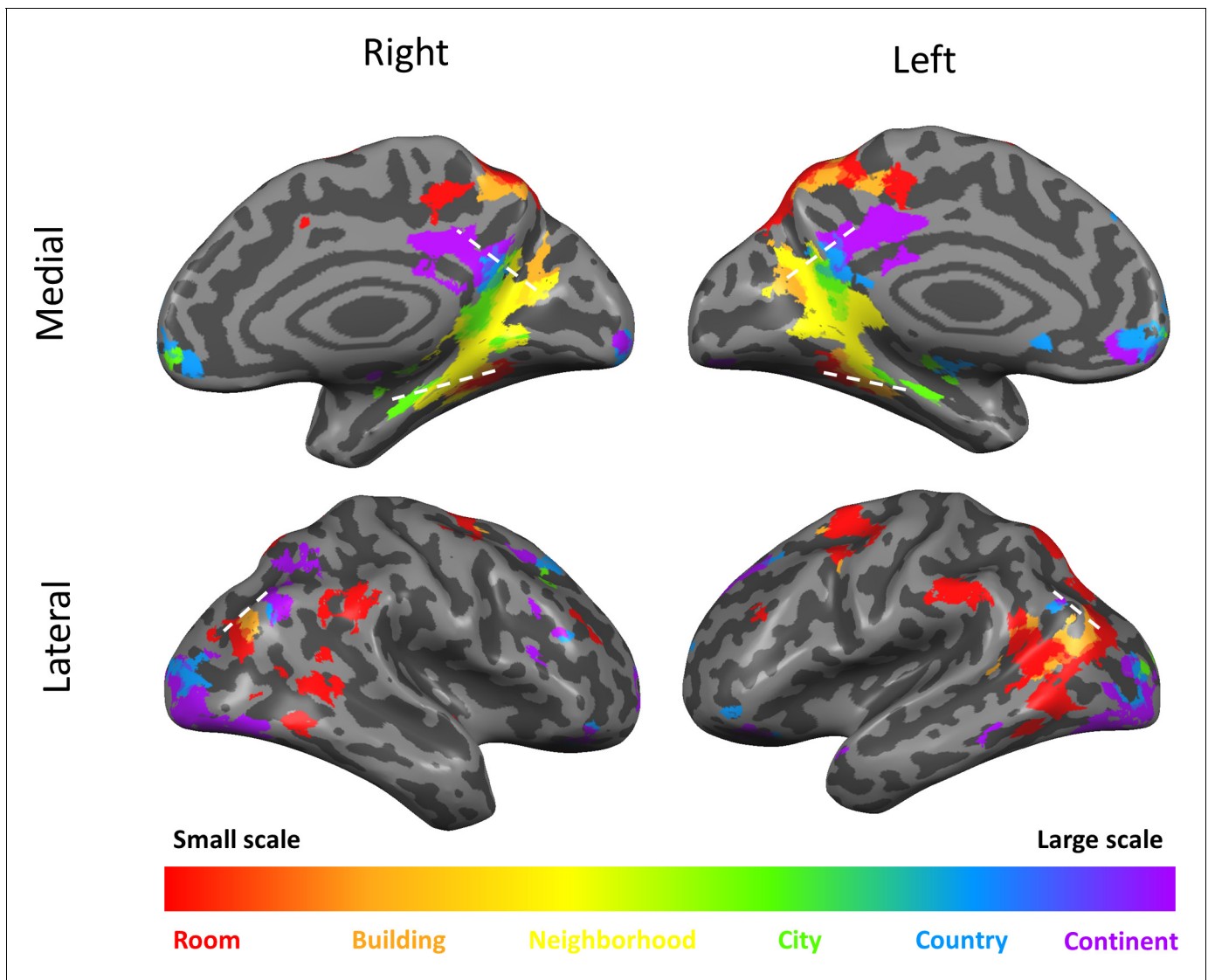

**Figure 3.** Scale-selective activity along gradients and additional cortical regions. Surface view of all scale-selective cortical activations (including regions outside of the three gradients; voxels identified by ANOVA across beta values, $p < 0.01$, FDR-corrected for multiple comparisons, minimum $r^2$ of fit = 0.7, cluster threshold = 15 mm$^2$). Continuous scale-sensitive gradients are marked by white dashed lines. Full volume maps of these results are available online at https://github.com/CompuNeuroPsychiatryLabEinKerem/publications_data/tree/master/spatial_scales (*Peer et al., 2019*, copy archived at https://github.com/elifesciences-publications/publications_data).

DOI: https://doi.org/10.7554/eLife.47492.011

with respect to our findings, we used masks of these regions as previously defined on an independent sample (*Julian et al., 2012*). The three regions (PPA, RSC and OPA) were found to be situated at the posterior part of the medial temporal, medial parietal and lateral occipito-parietal gradients, respectively. Accordingly, the scene-responsive regions were most active for the small and medium scales: room, building and neighborhood (*Figure 4*). This finding suggests their stronger involvement in the processing of immediate visible scenes, compared to more abstract larger environments. However, these regions also showed activity for the larger scales, suggesting that their computational role may extend beyond the exclusive processing of the immediately visible environment, though to a lesser extent (*Figure 4*).

### The three cortical gradients indicate a shift between the visual and default-mode brain networks

To relate the three cortical gradients to large-scale brain organization, we compared their anatomical distribution to a parcellation of the brain into seven cortical resting-state fMRI networks, as identified in data from 1000 subjects (*Yeo et al., 2011*). Across the three gradients, the posterior regions (related to processing of small scales) overlapped mainly with the visual network, while the anterior regions (related to processing of large scales) mainly overlapped with the default-mode network (*Supplementary file 1*).

### Differences in scale selectivity between the three cortical gradients

The previous analyses identified three cortical regions with gradual progression of scale selectivity. We next attempted to identify differences between these three regions that may be indicative of their functions. To this aim, we analyzed the number of voxels with preferential activity for each scale within each gradient (*Figure 5*, *Figure 5—figure supplement 1*). The medial parietal gradient was mostly active for the neighborhood, city and continent scales, indicating a role for this region in processing medium to large scale environments. In contrast, the medial temporal gradient contained mostly voxels sensitive to scales up to the city level, suggesting that this region is involved mostly in processing small to medium scales. Finally, the lateral occipito-parietal gradient was most active for the smallest scales (room, building) and the largest (continent) scale. These findings demonstrate that despite their similar posterior-anterior organization, the three scale-sensitive cortical gradients have different scale preferences, indicating possible different spatial processing functions.

### Subjects' behavioral ratings and their relation to the scale effects

Analysis of subjects' ratings of emotional significance and task difficulty for each location indicated no significant differences between scales, except for difficulty difference between the continent and the room and neighborhood scales (*Figure 1—figure supplement 2A–B*; correlation between difficulty and scale, r = 0.39; p<0.05, two-tailed one-sample t-test across subjects). Familiarity ratings did significantly differ across scales, with larger average familiarity for the smaller scale environments (*Figure 1—figure supplement 2C*; average correlation of familiarity and scale increase, r = −0.72; p<0.05, two-tailed one-sample t-test across subjects). First-person perspective taking and third-person perspective taking ratings were also highly correlated with scale increase, indicating a gradual shift between imagination of locations from a ground-level view in small-scale environments to imagination from a bird's-eye view in large-scale environments (r = −0.81, r = 0.80, respectively; both p<0.05, two-tailed one-sample t-test across subjects; *Figure 1—figure supplement 2E*, *Supplementary file 1*). Response times did not significantly differ between scales (*Figure 1—figure supplement 2D*). The verbal descriptions of task-solving strategy confirmed the trend of decrease in ground-level and increase in map-like (or 'bird's-eye') imagination with increasing scale (*Supplementary file 1*). These descriptions also demonstrated that as the scale decreased, subjects increasingly relied on estimations of walking or driving times between locations, except for the room scale where this strategy was not used (*Supplementary file 1*).

To measure the effect of these different factors on the observed activations, we used parametric modulation using subjects' ratings of emotion, familiarity, difficulty, perspective taking and strategy. The familiarity, perspective taking (first-person and third-person) and reports of use of a map strategy showed significant effects inside the scale-related gradients, in accordance with their high

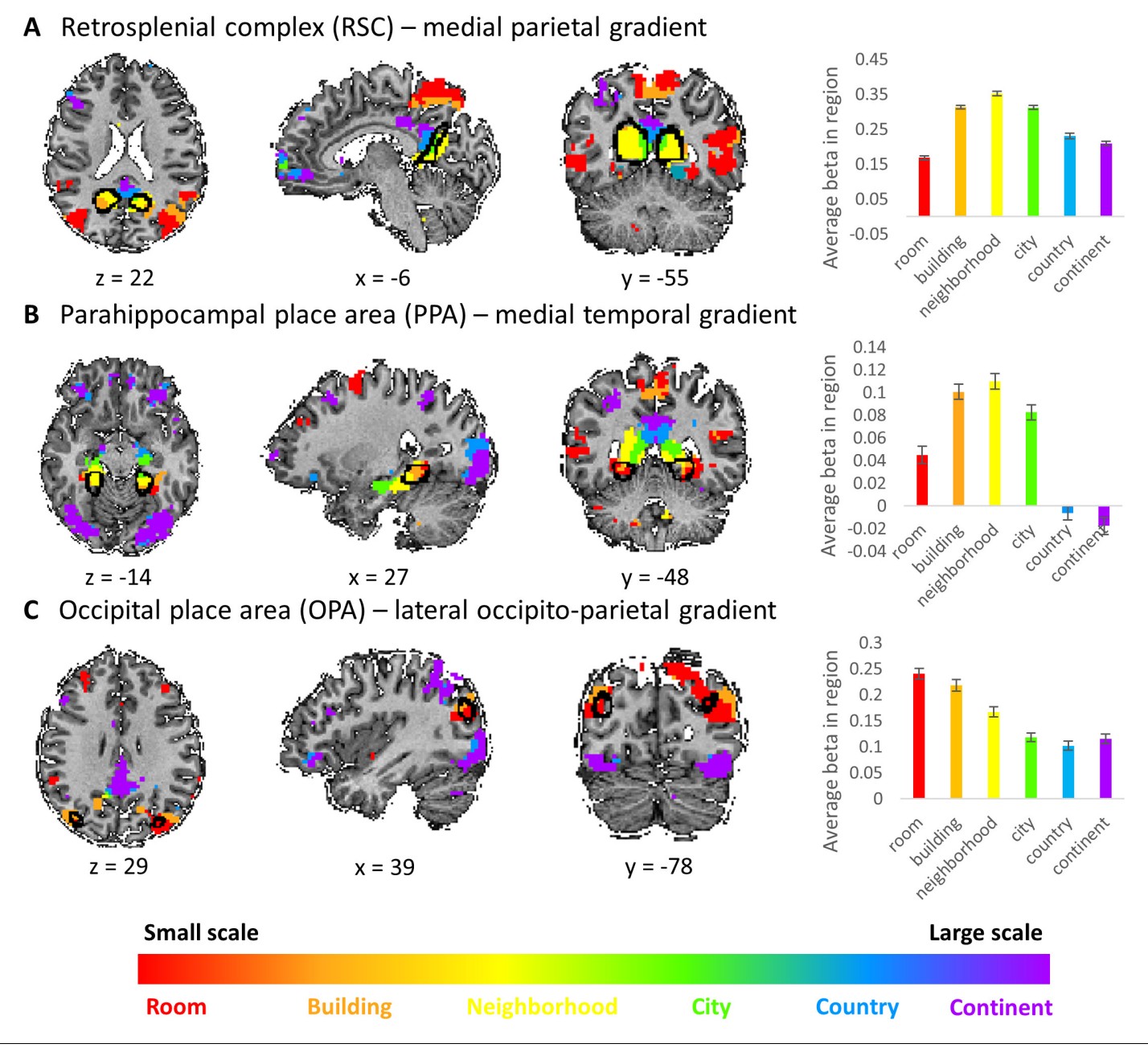

**Figure 4.** Visual scene-responsive cortical regions (PPA, RSC and OPA) are preferentially active for small to medium spatial scales. Scene-responsive cortical regions (marked by a black outline) were defined using publicly available dataset by responses to a places > objects contrast in a separate subject sample (*Julian et al., 2012*). (A) retrosplenial complex (RSC), (B) parahippocampal place area (PPA), (C) occipital place area (OPA). Left – overlap of scene-responsive regions and the three scale-sensitive gradients. Right– average beta weights for each condition (scale) within each region of interest (error bars represent standard errors across subjects). The visual scene-responsive regions are situated at the posterior part of the three gradients, and are therefore mostly active during processing of small to medium scale environments.

DOI: https://doi.org/10.7554/eLife.47492.012

correlation to spatial scale (*Figure 2—figure supplement 5*). No other factor showed any significantly active regions in this analysis.

We next contrasted the activity for the experimental task with that for the lexical control task at each region. Within the three gradients, this contrast revealed significantly higher activity for the spatial task compared to the lexical control task (GLM contrast, all p-values<0.05, FDR corrected for

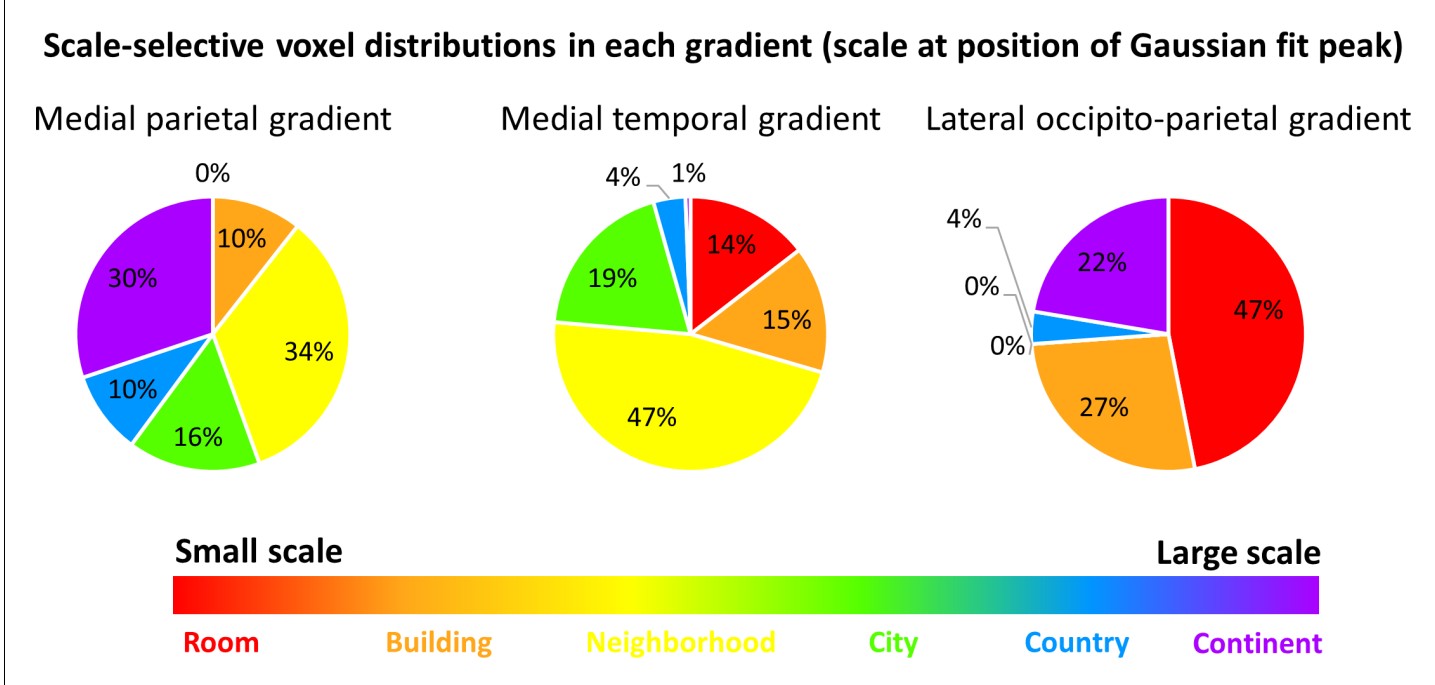

**Figure 5.** The three scale-selective cortical gradients have different voxel distributions, demonstrating preference for processing different spatial scales. The position of the Gaussian fit peak was used to identify voxels responsive to each scale. Voxel numbers are described within each gradient. Results indicate that the medial parietal gradient mostly represents scales at the neighborhood level and larger, the medial temporal gradient mostly represents environment up to the neighborhood-sized scales and has only small portions dedicated to larger scales, and the lateral occipito-parietal gradient is highly active both for the smallest scales and the largest ones.

DOI: https://doi.org/10.7554/eLife.47492.013

The following figure supplement is available for figure 5:

**Figure supplement 1.** Scale-selective voxel distributions within the three gradients.

DOI: https://doi.org/10.7554/eLife.47492.014

multiple comparisons across regions), except for the anterior city, country- and continent-related regions in the medial temporal gradient and the continent region in the occipito-parietal gradient. Among the other scale-sensitive regions outside of the gradients, only the supramarginal and lateral occipital cortex clusters did not show a significant activity above that of the lexical control task.

## Discussion

Our investigation revealed several novel findings. First, spatial scale sensitivity was found to be organized along three cortical gradients, extending anteriorly from the three known scene-responsive regions (PPA, RSC and OPA), as well as along the long axis of the hippocampus. These gradients were organized such that their posterior parts were most active for the smallest spatial scales and their anterior parts for the largest spatial scales. In addition, the posterior parts of the cortical gradients overlapped with the brain's visual network, while the anterior parts extended into the default-mode network (DMN). Spatial scale sensitivity was differentially distributed between these gradients, with the medial temporal gradient preferentially active for small- to medium-scale environments, the medial parietal gradient for medium- to large-scale environments, and the lateral occipito-parietal gradient for the small and large scales but not for medium-sized ones. These scale-selective gradients were correlated with a shift from detailed to less-detailed knowledge of locations, and from first- to third-person perspective taking with increasing scale. In the following, we discuss our results with respect to previous theories of spatial processing as well as findings regarding the spatial system's organization and its role in conceptual processing.

Several theories on how the cognitive system processes different spatial scales have been previously proposed. Early authors have suggested a scale-independent **unitary system** for spatial

representation, such as an hierarchical tree that stores relations between segments at each hierarchical level, irrespective of its scale (*Hirtle and Jonides, 1985*; *Holding, 1994*; *Worden, 1992*). In contrast, other authors have suggested that different neurocognitive system are responsible for representation of different spatial scales. According to **dual systems** theories, local room-sized environment are stored using a precise metric reference frame, and larger environments are represented as a schematic, non-metric graph connecting these smaller environments (*Meilinger, 2008*; *Werner et al., 2000*). Finally, **multiple systems** theories claim that separate systems process different spatial scales. The different suggested scales include figural/graphics spaces (smaller than the body), vista spaces (small environments that can be grasped from one location), navigation/environmental spaces (large spaces learned through navigation), and geographical spaces (regions too large to be learned by navigation, and are learned mainly through maps) (*Montello, 1993*; *Tversky, 2003*). Consequently, the different types of theories offer different predictions on the type of brain activity involved in computations at different scales. While dual and multiple systems theories would predict activation at different brain regions for different spatial scales, the unitary system theory would predict activity within same brain regions across scales. Our findings showed that all scale-sensitive regions are active across a range of spatial scales, with activity shifting along functional gradients within the same brain regions. Therefore, our findings seem to reconcile the different theories, showing a unitary system that is involved in spatial processing across a range of spatial scales, while nevertheless having an internal organization according to scale.

Several factors might explain the shift in cortical activity when subjects make judgments at different scales. One element that may differ between scales is the amount of movement involved in their navigation and initial learning, although we did not find consistent differences between reports of imagined movement at different scales. Alternatively, subjects may use personally-relevant episodic memories to a different degree in order to perform the task at different scales, although the limited time allowed for each comparison and the lack of differences in emotional significance ratings between locations limit this possibility. Other potential contributing factors include differences in the level of familiarity/detailed knowledge of locations between the different scales, and a shift in perspective taking between first- and third-person imagination. Subjects' behavioral ratings and verbal descriptions show that when thinking of larger scales, subjects shift to use a bird's-eye imaginary perspective and have less detailed knowledge of locations. Previous studies that directly manipulated familiarity and level of knowledge of locations (*Epstein et al., 2007b*; *Epstein et al., 2007a*; *Hirshhorn et al., 2012b*; *Wolbers and Büchel, 2005*) or first- vs. third-person perspective taking (*Rosenbaum et al., 2004*; *Sherrill et al., 2013*) found differences in activity within the OPA, RSC and PPA (generally higher activity for more well-known places and first-person perspective, as found here). However, these studies did not find activity shift to more anterior cortical regions for third-person perspective taking or less well-known locations, as shown in the scale-sensitive gradients described in this study. Therefore, level of familiarity and perspective taking might not entirely explain the observed gradients. These findings might be explained by the idea that posterior gradient regions contain detailed spatial information, supported by the visual system and acquired using a first-person perspective; as the scale increases, knowledge becomes less detailed and more abstract and schematic, supporting a bird's-eye/map like imagination (*Arzy and Schacter, 2019*).

Past studies have found evidence for posterior-anterior subdivisions in PPA and RSC (*Baldassano et al., 2016*; *Baldassano et al., 2013*; *Burles et al., 2018*; *Chrastil et al., 2018*; *Silson et al., 2019*; *Silson et al., 2016*). Posterior parts of both regions were active during visual scene viewing and navigation, and were functionally connected to visual regions. In contrast, anterior regions were active during imagination and recall of relations between unseen parts of the larger environment, and were functionally connected to the anterior hippocampus and DMN. These findings were interpreted as evidence for two spatial systems: a posterior system involved in perceptual analysis and encoding of visual scenes, and an anterior system responsible for scene recall from memory (*Baldassano et al., 2016*; *Burles et al., 2018*; *Chrastil et al., 2018*). Our results provide several novel insights related to these past findings. First, instead of a binary distinction between two systems in scene-selective regions (PPA, RSC and OPA), we found a continuous gradient operating both within these regions and extending anteriorly from them. Second, all investigated conditions involved only recall of environments from memory, suggesting that posterior-anterior activity differences do not relate only to direct visual processing vs. scene memory. Instead, the scale of representation may be important for organizing activity along the posterior-anterior axis. Third, we

found that the cortical posterior-anterior organization by spatial scale also exists along the hippocampal long axis, in agreement with past findings (*Brunec et al., 2018*; *Kjelstrup et al., 2008*). With respect to the hippocampus, hippocampal long axis organization was previously suggested to relate to the level of detail vs. abstractness of the representation, both in space and in other memory domains (*Brunec et al., 2018*; *Poppenk et al., 2013*). We hypothesize that the same principle of detailed vs. general-schematic representation applies to the scale-sensitive cortical gradients we identified. Indeed, behavioral works demonstrated that while humans represent small-scale environments in a precise, Euclidean manner, in larger environments they may be using a more flexible representation system (*Meilinger, 2008*; *Wolbers and Wiener, 2014*). This representation may take the form of a 'cognitive graph' that represents relations between locations topologically (*Chrastil and Warren, 2014*; *Epstein, 2008*; *Meilinger, 2008*; *Warren et al., 2017*), resulting in behavioral biases and navigational mistakes (*Moar and Bower, 1983*; *Tversky, 1981*). Thus, the general posterior-anterior organization of the spatial system may relate to precise relational encoding in posterior regions vs. a flexible, cognitive-graph-like representation of larger spaces in anterior regions.

Across the three cortical gradients, we found that posterior regions correspond to visual scene-processing regions (PPA, RSC and OPA), while anterior regions were part of the default-mode network (DMN), in accordance with previous findings (*Baldassano et al., 2016*; *Baldassano et al., 2013*; *Chrastil et al., 2018*). The RSC, PPA and OPA are considered to be regions specializing in spatial processing (*Dilks et al., 2013*; *Epstein and Kanwisher, 1998*; *Epstein and Ward, 2010*). In contrast, the DMN is active both during rest and across a variety of high-level, mostly self-referenced, cognitive processes, that extend beyond the spatial domain (*Buckner et al., 2008*; *Buckner and Carroll, 2007*; *Peer et al., 2015*; *Simony et al., 2016*; *Spreng et al., 2009*). Thus, the posterior-anterior gradients we identified might reflect a shift from representing visually observable spatial relations in small-scale spaces to representing more abstract relations in larger environments. Indeed, recent investigations suggested a general cortical organization scheme, where information gradually progresses from sensory regions to form high-level cognitive representations in the DMN (*Huntenburg et al., 2018*; *Margulies et al., 2016*). In a previous study, we found that posterior regions within medial parietal cortex specialize in processing spatial relations, while the regions anterior to them process temporal relations between events and social relations between people (*Peer et al., 2015*). Similarly, it has been shown that posterior RSC and hippocampus are more active for spatial judgments while regions anterior to them are active for general episodic memory (*Hirshhorn et al., 2012a*). Moreover, a posterior-anterior gradient was observed in studies of brain processing of different scales of time, when transitioning from small, immediately-perceivable temporal windows (single seconds) to larger windows (several minutes) that require integration across time (*Baldassano et al., 2017*; *Chen et al., 2016*; *Hasson et al., 2008*). The hippocampus and its posterior-anterior organization were also related in previous works to processing of both spatial and non-spatial knowledge (*Eichenbaum, 2000*), leading to suggestions that representation of conceptual knowledge relies on a geometric, spatial-like processing system (*Behrens et al., 2018*; *Bellmund et al., 2018*; *Casasanto and Boroditsky, 2008*; *Gärdenfors, 2000*; *Liberman and Trope, 2008*; *Parkinson and Wheatley, 2013*). Our findings suggest that also outside the hippocampus, the scene-selective RSC, PPA and OPA, which are usually studied in isolation within the field of spatial neuroscience, may combine with the DMN to form a generalized brain system for conceptual knowledge organization.

Besides the three gradients, several other bilateral cortical regions showed sensitivity to spatial scale. These regions included the superior frontal gyrus, supramarginal gyrus, posterior temporal cortex and dorsal precuneus, which had the highest activity for the smallest spatial scale (room) and decreased activity with increasing scales. These regions may be involved in processes that are preferentially involved in analysis of local environments, such as egocentric perspective taking, in accordance with our subjects' reports (*Figure 1—figure supplement 2*, *Supplementary file 1*) and the parietal cortex's role in egocentric processing of the immediately surrounding environment (*Byrne et al., 2007*; *Wilson et al., 2005*). In contrast, clusters at the lateral occipital and medial prefrontal cortices displayed the opposite pattern of maximal activity at large spatial scales and decreasing activity with decreasing scale. This pattern may reflect processes that are employed more at large scales, such as visualization of maps and routes that occurs when imagining large-scale spaces (*Montello, 1993*; *Tversky, 2003*), in accordance with subjects' reports (*Supplementary file*

*1*). Future experiments may untangle the role of each of these activation clusters in small-scale and large-scale specific processing.

Our findings offer several new insights regarding the distinct roles of different parts of the brain's spatial processing system. The medial parietal gradient, extending from the RSC, was found to be preferentially active for processing of large environments, ranging from neighborhoods to continents. Previous research has shown that the RSC is involved in locating places within their large-scale context, such as when pointing in the direction of far-away, unseen landmarks (*Epstein, 2008*; *Epstein et al., 2007b*; *Maguire, 2001*). Additionally, it was suggested to be related to representations of approximate relations between locations as a cognitive graph (*Epstein, 2008*; *Epstein and Vass, 2014*). Therefore, the RSC may be involved in locating places within their larger context across environments of different sizes. Interestingly, a recent meta-analysis demonstrated that the posterior part of the RSC/posterior cingulate cortex is active when directly viewing scenes while its more dorso-anterior part is active when locating items in larger unobservable environments, further supporting this interpretation and the gradients we identified (*Burles et al., 2018*). In a similar manner, the medial temporal gradient, extending from the PPA, was shown here to be responsive mostly for environments up to the neighborhood level. The PPA is classically known to be involved in location recognition and analysis of observed scene layouts (*Epstein, 2008*). Our findings suggest a role for the PPA in performing similar computations not only in directly visible scenes, but also in larger environments that can still be learned by experience. Finally, the lateral occipito-parietal gradient, extending from the OPA, was shown here to be primarily involved in processing room to building sized environments, but also to have an anterior extension involved in processing very large environments. The OPA is thought to be a perceptual processing system used for analyzing local geometry and identifying routes within visual scenes (*Bonner and Epstein, 2018*; *Bonner and Epstein, 2017*), and our findings suggest it may have similar functions in very large-scale spaces, possibly due to human tendency to visually imagine these scales as maps. Taken together, the anterior extension of the PPA, RSC and OPA suggest that they perform general computations across different environment sizes, beyond the immediately-visible scenes by which they are usually defined.

When looking at the overall distribution of spatial scale selective voxels across the brain, it is apparent that some spatial scales are more prominently represented than others. The smallest environments (rooms) were preferentially represented along large parts of the lateral parieto-temporal cortex, indicating their importance in everyday experience and behavior of the environment immediately surrounding us. Among the medium scales, neighborhoods showed the largest prominence and largest number of maximally active voxels along the three gradients. Regions in the size of neighborhoods may be the most directly relevant for everyday active navigation and social communication; indeed, most monkeys and apes have a territory size of up to 3 km$^2$ (*Lowen and Dunbar, 1994*), suggesting that this is the scale that has been most relevant to navigation in primate (and possibly human) evolution. Finally, several regions in the lateral occipital and medial prefrontal lobes, as well as in the anterior parts of the three cortical gradients, showed prominent activity specifically at the largest spatial scale of continents. This specificity may be related to the increased abstractness of relations as perceived in these large environments, or to the use of mechanisms specifically involved in imagining very large spaces, such as their conception through maps (*Montello, 1993*; *Tversky, 2003*).

Activity in the hippocampus, and in some of the anterior parts of the cortical gradients, was negative relative to baseline, while showing consistent differences in activity between scales. The anterior parts of the three cortical gradients overlap with the DMN, which may be characterized by negative BOLD during tasks (*Raichle et al., 2001*), and negative BOLD in the hippocampus is also a common finding (*Shipman and Astur, 2008*). These negative activations were interpreted in the past as potentially reflecting high constitutive activity of these regions during rest more than during active tasks (*Ekstrom, 2010*; *Shipman and Astur, 2008*; *Stark and Squire, 2001*). The fact that these activations are below baseline preclude inference of whether these regions participate in processing of smaller spatial scales or are only active for larger ones.

Our study has several limitations. First, the task we used involved a specific cognitive computation of three-way distance comparison between locations, enabling direct comparison between scales using the same task and experimental design. However, experiments involving other tasks that can be applied across spatial scales may reveal additional information on scale-specific and scale-independent brain processes. Second, to obtain a large range of spatial scales and maintain ecological

validity we used a personalized paradigm where subjects provided names of real-world locations familiar to them, in six naturalistic scales, therefore not controlling for the precise size and distances in each scale. Despite this restriction, the distances between subjects' selected stimuli logarithmically increased with each scale, and a bilateral gradient organization was consistently observed across gradients. However, the exact relations between distances and scales may be further investigated in a more granular manner using studies of well-controlled (e.g. virtual) environments with different scales. Third, to identify the DMN and the known scene-selective cortical regions, we used group averages from large subject groups; directly identifying these systems at the single-subject level might yield more detailed measurements of their scale specificity. Fourth, we did not measure navigation, imagery or memory abilities, and therefore did not control for these factors in the group analyses; however, our results hold at the individual subject level in the large majority of subjects, limiting their ability to explain our results. Finally, as discussed above, environments at different scales may have inherent differences in their imagination and the strategies employed for judgments within them, such as imagination of walking, driving, flying or imagining them through maps. Although we cannot rule out these factors as affecting activation differences between scales, a shift between different strategies is not likely to explain a continuous shift in the location of activity along cortical gradients with a change in spatial scale, as we observed here.

In conclusion, our results demonstrate the extension of known visual scene-responsive regions to a larger scheme of brain organization and processing of relations in larger unseen environments. These findings may provide a basis for understanding how the human brain processes and integrates the navigated environment across scales. Furthermore, our findings suggest a way by which brain systems responsible for representation of large-scale environments may be used to flexibly represent information in other abstract cognitive domains. Further investigations into how the brain integrates environments and relations in large scales may inform us on general processing mechanisms in the brain, and how relations in other abstract conceptual domains are encoded by spatially-based processes.

## Materials and methods

### Subjects
Nineteen healthy subjects (twelve males, mean age 27.7 ± 4.4 y) participated in the study. All subjects provided written informed consent, and the study was approved by the ethical committee of the Hadassah Hebrew University Medical Center.

### Experimental stimuli
Six spatial scales were investigated: room, building, neighborhood, city, country and continent. These scales reflect ecological categories, which grow in size in a logarithmic manner (*Figure 1—figure supplement 1*). To gather stimuli for this large range of spatial scales, subjects were asked to provide names of two real-world locations personally familiar to them at each scale, several days before the experiment (e.g. home bedroom, Hadassah hospital, London, Argentina). In each of these twelve locations, subjects indicated the names of eight items whose location they personally know: objects at the room and building scale (e.g. bed, vending machine), and landmarks at the neighborhood, city, country and continent scales (e.g. Supermarket, Eiffel tower). Subjects were asked to keep the item names short and make sure they represent a unique location. Subjects who failed to provide enough personally-familiar stimuli (due to lack of sufficient travel experience abroad) were not included in the experiment.

### Experimental paradigm
During the experiment, subjects were presented with a target stimulus consisting of one of the items they had provided and its respective location (e.g. 'table' in 'living room', 'city hall' in 'Jerusalem'), followed by a pair of other stimuli from the same location on the left and right of the screen (*Figure 1*). Subjects were instructed to indicate which of the two stimuli is closer to the target stimulus by pressing the left or right buttons.

Stimuli were presented in a randomized block design. Each block started by presentation of a target stimulus for 2.5 s, followed by consecutive presentation of four stimuli pairs, each for 2.5 s

(*Figure 1*). All stimuli within the same block had to be judged in relation to the block's target stimulus location. Each block (12.5 s) was followed by 7.5 s of fixation. Subjects were instructed to respond accurately but as fast as possible. The experiment consisted of either four or five experimental runs for each subject, each run containing 24 blocks in a randomized order (two blocks for each of the twelve locations = four blocks in each spatial scale). In total, subjects performed 24 blocks per run, each including four object pairs, for a total of 384–480 comparisons over the experiment. Anchor items and stimuli pairs were chosen independently and randomly from the eight items the subject provided for each location, allowing for repetitions; on average, 3.5% of stimuli pairs were repeated during the experiment (with the same anchor stimulus), and each item was used $9 \pm 2.87$ times as a target. In addition, eleven subjects performed a lexical control task in a separate run, in which they viewed similar target stimuli followed by stimuli pairs but were instructed to indicate which of the pair of words is closer in length to the target stimulus. A training task using pairs of stimuli derived from the same pool was delivered before the experiment; subjects performed the training until they indicated that they felt comfortable doing the task (average number of training trials per subject = $53 \pm 26.6$, or 8.8 trials per spatial scale). Stimuli were presented using the Presentation software (Version 18.3, Neurobehavioral Systems, Inc, Berkeley, CA, www.neurobs.com, RRID: SCR_002521). After the experiment, subjects rated their level of familiarity with each of the twelve locations, the emotional significance of the location, and level of difficulty of judgments at each location (from 1 to 7). They were also asked to describe the strategy used for determining responses in each of the six spatial scales (free descriptions) and specifically to what extent did they adopt a ground-level or bird's-eye point-of-view (1 to 7 rating).

## Analysis of spatial scale sizes

For each stimulus provided by each participant, we identified the latitude and longitude of the stimulus location, if it was a name which could be identified. 72% of stimuli locations were identified (65% for neighborhoods, 83% for cities, 72% for countries and 70% for continents). The pairwise distances between all items in each location and scale were calculated using the Haversine formula (to account for the earth's globular shape), using a script provided by M Sohrabinia: https://www.math-works.com/matlabcentral/fileexchange/38812-latlon-distance. A linear fit to the resulting logarithmic values shows a fit of $r^2 = 0.98$, indicating that scale transitions reflect a logarithmic increase in environmental size.

## MRI acquisition

Subjects were scanned in a 3T Siemens Skyra MRI (Siemens, Erlangen, Germany) at the Edmund and Lily Safra Center (ELSC) neuroimaging unit. Blood oxygenation level-dependent (BOLD) contrast was obtained with an echo-planar imaging sequence [repetition time (TR), 2,500 ms; echo time (TE), 30 ms; flip angle, 75°; field of view, 192 mm; matrix size, 64 × 64; functional voxel size, 3 × 3 × 3 mm; 46 slices, descending acquisition order, no gap; 200 TRs per run]. In addition, T1-weighted high resolution (1 × 1 × 1 mm, 160 slices) anatomical images were acquired for each subject using the MPRAGE protocol [TR, 2,300 ms; TE, 2.98 ms; flip angle, 9°; field of view, 256 mm].

## MRI processing

fMRI data were processed and analyzed using the BrainVoyager 20.6 software package (R. Goebel, Brain Innovation, Maastricht, The Netherlands, RRID:SCR_013057), Neuroelf v1.1 (www.neuroelf.net, RRID:SCR_014147), and in-house Matlab (Mathworks, version 2018a, RRID:SCR_001622) scripts. Preprocessing of functional scans included slice timing correction (cubic spline interpolation), 3D motion correction by realignment to the first run image (trilinear detection and sinc interpolation), high-pass filtering (up to two cycles), smoothing (full width at half maximum (FWHM) = 4 mm), exclusion of voxels below intensity values of 100, and co-registration to the anatomical T1 images. Anatomical brain images were corrected for signal inhomogeneity and skull-stripped. All images were subsequently normalized to Montreal Neurological Institute (MNI) space (3 × 3×3 mm functional resolution, trilinear interpolation). The full analysis and preprocessing scripts are available at https://github.com/CompuNeuroPsychiatryLabEinKerem/publications_data/tree/master/spatial_scales (*Peer et al., 2019*, copy archived at https://github.com/elifesciences-publications/publications_data).

## Functional MRI analysis

### Estimation of cortical responses to each spatial scale

A general linear model (GLM) analysis (*Friston et al., 1994*) was applied at each voxel, where predictors corresponded to the six spatial scales. Each modeled predictor included all experimental blocks at one spatial scale, where each block was modeled as a boxcar function encompassing the target stimulus and the four distance comparisons following it. Predictors were convolved with a canonical hemodynamic response function, and the model was fitted to the BOLD time-course at each voxel. Motion parameters were added to the GLM to eliminate motion-related noise. In addition, white matter and CSF masks were manually extracted in BrainVoyager for each subject (intensity >150 for the white-matter mask and intensity <10 with a bounding box around the lateral ventricles for CSF), and the average signals from these masks were added to the GLM to eliminate potential noise sources. Data were corrected for serial correlations using the AR(2) model and transformed to units of percent signal change. Subsequently, a random-effects analysis was performed across all subjects to obtain group-level beta values for each predictor.

### Identification of voxels with spatial scale sensitive activity

To identify voxels with differences in brain activity between spatial scales, single-factor repeated-measures ANOVA was applied in each voxel on the scale-specific predictors' beta values, across all subjects (FDR-corrected for multiple comparisons across voxels, p<0.01). Following voxel identification, beta values were averaged for each voxel across subjects, and two methods were used to identify selectivity to spatial scales: (1) fitting a Gaussian function to the betas' graph and identifying its peak; (2) selection of the scale with maximal activity (*Figure 2—figure supplement 1*). Since the responses in almost all regions follow a gradual pattern of change between different scales, the Gaussian fit enables a fuller consideration of the overall pattern of activity and scale selectivity across scales, instead of focusing only on the maximally active scale. Gaussian fitting was performed for each beta vector after its normalization by subtracting its minimum value, and fitting was performed using Matlab, with bounds of 0 to 100 for amplitude, −100 to 100 for center, and 0 to 100 for width. Only voxels with fit of $r^2$ >0.7 (5737 out of 7452 voxels) were included in the subsequent analyses of Gaussian fit peaks.

### Group-level analysis of activity profiles across spatial scales

Event-related activity (ERA) averaging and beta averaging across subjects were used to investigate activity profiles at each region. For event-related activity, BOLD signals were averaged for all blocks containing each scale across all runs and subjects, for the ten functional volumes following each block's initial display of the target stimulus. Beta plots were also created by averaging the beta values calculated in the random-effects GLM analysis across all subjects. These procedures were performed in each region of interest, as defined by the peak of the Gaussian fit to the group-averaged beta maps.

### Measuring increase of scale selectivity along gradients and along the hippocampal long axis

Within the hippocampus and the three identified gradients (medial temporal, medial parietal and lateral occipito-parietal), peak of Gaussian fit was averaged for each MNI coordinate along the Y axis, as well as scale with maximal response, resulting in vectors of scale selectivity across the posterior-anterior axis. To measure whether there is a gradual increase in preferred scale along each gradient, we modeled each gradient using a linear function that was fitted to the scale preference values along it, and also fitted this function to 1000 shuffled versions of each scale preference vector for obtaining a null distribution. The slope of the actual fit was tested against the slope of the fits to the random permutations to check if the obtained increase in scale preference along the gradients significantly deviates from chance. Resulting p-values were corrected for multiple comparisons across gradients using the false discovery rate (*Benjamini and Hochberg, 1995*). This analysis was additionally repeated at the individual subject level, by fitting the Gaussian function at the individual subject level and calculating the linear fit and its significance along the cortical gradients (regions defined by the group results) and along the hippocampus.

## Comparison to hippocampus and visual scene-responsive regions (RSC, PPA and OPA)

Masks of the RSC, PPA and OPA were used, as established in a previous publication (*Julian et al., 2012*; http://web.mit.edu/bcs/nklab/GSS.shtml). These masks represent group activation clusters from 30 subjects who watched visual images with a contrast of scenes > objects. The outlines of the group-level clusters were overlaid on each cortical gradient (*Figure 4*) to compare their cortical locations. In addition, a region-of-interest GLM analysis (random effects group analysis) was performed within each mask, to obtain beta values for each spatial scale at each region. The hippocampal region-of-interest was extracted from the Harvard-Oxford atlas brain template distributed with FSL (http://www.fmrib.ox.ac.uk/fsl/, RRID:SCR_001476; *Desikan et al., 2006*; *Jenkinson et al., 2012*).

## Comparison of scale-specific activations to large-scale resting-state networks

A previously published whole-brain parcellation into seven large-scale brain networks was used as a template for resting-state networks location. For each scale-selective region within the three gradients, its percent of overlap with each of the seven resting-state networks was measured (percent of voxels from *Desikan et al., 2006* this region within each network).

## Analyses of potential factors contributing to the scale effect

Each subject's ratings of difficulty, emotional significance and familiarity for each location were independently normalized by z-transform. Ratings of first-person perspective taking, third-person perspective taking, and mentions of use of different strategies were similarly transformed for each scale. The resulting values were then used as parametrically modulation regressors (after convolution with the hemodynamic response function), according to each experimental block's spatial scale and specific location. A response time predictor was added in a similar manner according to each trial's response time. Random-effects group analysis (corrected for serial correlations, AR(2)) was then performed using each of the regressors separately, to identify activity modulation by each potential contributing factor. In addition, one-way ANOVA (Tukey-Kramer post-hoc test, $p < 0.01$) was used to identify significant differences in the ratings between the six spatial scales.

## Comparison of activity to the lexical control task

Regressors for the lexical control were added to the scale predictors in the GLM analysis, and a new design matrix was computed for each subject. A group analysis (corrected for serial correlations, AR (2)) was performed in each scale-sensitive region of interest, and activity in this region's preferred scale was contrasted with the activity corresponding to the respective control condition.

### Data sharing

All of the analysis codes from this project, as well as the resulting statistical maps and spatial scale-specific regions, are freely available at https://github.com/CompuNeuroPsychiatryLabEinKerem/publications_data/tree/master/spatial_scales (*Peer et al., 2019*, copy archived at https://github.com/elifesciences-publications/publications_data).

## Acknowledgements

This work was supported by the Israeli Science Foundation (Grant No. 1306/18 and 3213/19). MP is supported by a Fulbright postdoctoral fellowship from the United States–Israel Educational Foundation, and by the Eva, Luis and Sergio Lamas Scholarship Fund. We wish to thank our study participants, Assaf Yohalashet, Yuval Porat, Lee Ashkenazi and Leon Deouell from the ELSC neuroimaging unit for their help in MRI scanning, Noam Saadon-Grosman for help with the analyses, and Gregory Peters-Founshtein and Rachel Fried for helpful comments.

## Additional information

### Funding

| Funder | Grant reference number | Author |
|---|---|---|
| Israel Science Foundation | 1306/18 and 3213/19 | Shahar Arzy |
| Fulbright Association | | Michael Peer |
| Eva, Luis and Sergio Lamas Scholarship Fund | | Michael Peer |

The funders had no role in study design, data collection and interpretation, or the decision to submit the work for publication.

### Author contributions

Michael Peer, Conceptualization, Software, Formal analysis, Validation, Investigation, Visualization, Methodology, Writing—original draft, Writing—review and editing; Yorai Ron, Rotem Monsa, Software, Formal analysis, Validation, Investigation, Methodology, Writing—original draft; Shahar Arzy, Conceptualization, Software, Formal analysis, Supervision, Funding acquisition, Validation, Investigation, Methodology, Writing—original draft, Writing—review and editing

### Author ORCIDs

Michael Peer (ID) https://orcid.org/0000-0002-8373-8558
Shahar Arzy (ID) https://orcid.org/0000-0001-6500-8095

### Ethics

Human subjects: All subjects provided written informed consent, and the study was approved by the ethical committee of the Hadassah Hebrew University Medical Center (protocol 0657-15-HMO).

### Decision letter and Author response

Decision letter https://doi.org/10.7554/eLife.47492.022
Author response https://doi.org/10.7554/eLife.47492.023

## Additional files

### Supplementary files

• Supplementary file 1. Supplementary tables. Table S1: coordinates of all scene-sensitive activations, sorted by the spatial scale at the position of the Gaussian fit peak. Table S2: relation of cortical gradients to large-scale resting-state brain systems. Table S3: verbal descriptions of strategy used in task performance for each spatial scale.
DOI: https://doi.org/10.7554/eLife.47492.015

• Transparent reporting form
DOI: https://doi.org/10.7554/eLife.47492.016

### Data availability

All of the statistical maps and analysis codes are available at the GitHub repository: https://github.com/CompuNeuroPsychiatryLabEinKerem/publications_data/tree/master/spatial_scales (copy archived at https://github.com/elifesciences-publications/publications_data).

The following previously published datasets were used:

| Author(s) | Year | Dataset title | Dataset URL | Database and Identifier |
|---|---|---|---|---|
| Yeo BT, Krienen FM, Sepulcre J, Sabuncu MR, Lashkari D, Hollinshead M, Roffman JL, | 2011 | The organization of the human cerebral cortex estimated by intrinsic functional connectivity | https://surfer.nmr.mgh.harvard.edu/fswiki/CorticalParcellation_Yeo2011 | FreeSurfer, CorticalParcellation_Yeo2011 |

Smoller JW, Zollei L., Polimeni JR, Fischl B, Liu H, Buckner RL

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
