## [Decision Letter]

Thank you for submitting your article "Processing of different spatial scales in the human brain" for consideration by *eLife*. Your article has been reviewed by three peer reviewers, and the evaluation has been overseen by a Reviewing Editor and Timothy Behrens as the Senior Editor. The following individuals involved in review of your submission have agreed to reveal their identity: Dori Derdikman (Reviewer #2); Buddhika Bellana (Reviewer #3).

The reviewers have discussed the reviews with one another and the Reviewing Editor has drafted this decision to help you prepare a revised submission.

Summary:

This study explores the representation of scale-dependent spatial information ranging from small to large geographical spaces and from concrete to abstract features. Using a novel fMRI task, human participants were asked to perform judgments of spatial distance (i.e., which of the two following items are closer to a cued item), where the spatial scale of the distance judgment is manipulated within participants. Scale-dependent activity was evident in three posterior-anterior gradients (medial-temporal cortex, medial parietal cortex and lateral parieto-occipital cortex). Within each of these regions, a significant linear fit was observed, such that more anterior voxels were preferentially recruited during judgments of larger spatial scales (e.g., city, country, continent), while posterior voxels were associated with judgments at more local scales (e.g., room, building). The findings reported here complement and extend previous work in both rodents and humans, which have mostly focused on smaller-scale environments and scenes.

Essential revisions:

Task Design:

1) The task description requires further detail to understand exactly what was required of participants and, in turn, how to interpret the results. It would be useful to share the specific decisions each participant had to make for each level of spatial scale, or at the very least, examples from each level of spatial scale. An example at the room level is given, but similar in-text samples of at least one block per spatial scale is necessary for the reader to have a better handle on the actual decisions the participants made. Consider adding a short description of the task before the results, and ideally a Methods figure in the main text accompanied by a set of example cues and trials from each of the 6 spatial scales.

2) While the increasing levels of spatial scale make intuitive sense, the cut-off between each category seems rather arbitrarily defined. For example, it was not clear why the spatial scales progressed from house to neighbourhood when an interim level of street could be made. Similarly, the leap from city to country further seemed to neglect intermediate spatial scales such as region/county etc. It would be helpful for the authors to acknowledge or justify the use of these seemingly arbitrary cut-off points between these spatial scales and to perhaps consider how a more granular classification system might result in different findings.

3) The specific composition of object pairs in the distance judgments task is not clearly presented. From my understanding, participants provided 2 locations per level of spatial scale (total = 12 locations), and then produced 8 relevant items each. Then the task itself had 4-5 runs that contained 4 blocks per spatial scale (total = 16 blocks). Each block then contained 4 stimulus pairs and a target/cue, anchoring the participants' decisions. Therefore, there should be at least 4*4*4 (# of runs * # of blocks * # of pairs) pairs of items for each spatial scale across the experiment.

4) Were all pairs unique for each spatial scale and subject, or were there repetitions?

5) Were all items drawn exclusively from combinations of the 12 items produced by the corresponding subject?

6) How was the cued item (e.g., "the bed") selected per block?

7) Were all 12 items produced by each subject used as targets? In pairs? Were they all presented an equal number of times?

8) Were these 12 same items used in the 2-5 minutes training task?

9) On how many runs of each spatial scale were the participants trained? Did each training run have the same parameters as the experimental run (e.g., number of trials/object pairs)?

10) Were the post-task difficulty judgments done for each specific object pairing, or was a rating produced per individual object? E.g.1-7 difficulty rating for "Table - Window"? 1-7 difficult rating for "Table"?

Neuroimaging data:

11) What precisely was the modelled predictor? The specific onset of each object pair presentation from each block, per spatial scale, drawn across runs (e.g., stick function)? Or was it a boxcar function that modelled activity for the duration of each block? Please clarify.

12) Was there any additional nuisance regression that was conducted (e.g., white matter, or cerebrospinal fluid)? If not, why?

13) In terms of analysis, were trends other than linear ever tested? Did the linear fit have the most explanatory power relative to other potential trends (e.g., quadratic, cubic …)?

14) The behavioral data (Figure 1—figure supplement 2 and Supplementary file 3) should be used to segment the fMRI data, as at least in some cases it could provide an additional explanation to some of the gradients. Specifically:a) When grouping data according to the categories in Supplementary file 3, what does the fMRI signal look like? (e.g. imagining a map-like view vs. triangulation).b) How does the fMRI signal segment when weighing according to Familiarity rating, or Perspective-taking rating? (Figure 1—figure supplement 2)

Interpretation:

15) A fundamental question relates to the nature of smaller versus larger scale environments. For example, a room is far less dynamic than a city and does not necessarily require any movement or navigation within the spatial array to correctly adjudicate between the two distances. How can we determine whether scale preference is the critical factor versus the type of experience that one has at these different spatial scales? Participants were not asked whether the judgments within smaller scale environments invoked retrieval of specific prior experiences (i.e., episodic memory). Medial temporal and medial parietal activation for small-to-large environments may not necessarily reflect the spatial scale but the harnessing of personally relevant episodic experiences during the task. Please discuss.

16) The authors note the recruitment of the DMN for larger spatial scales and suggest the potential relationship with representations of abstract domains more generally. It may be interesting to examine this directly using a tool like Neurosynth (http://neurosynth.org). For example, one could examine the% voxel overlap between whole brain maps at each spatial scale and meta-analytic maps of "abstract" and "concrete" obtained via Neurosynth.

17) In Figure 1 (E, bottom right panel), the hippocampal long axis does not show a clear scale-dependent activity gradient, contrary to the claim in the Discussion section, and to previous works (subsection “Three posterior-anterior gradients of spatial scale selectivity”). Activity spans three scales (neighborhood->country) but seems to be involved mainly with 'city' scale. Furthermore, surprisingly it is not selective at all in 'room'/'building' scale. The above have been well established in both VR (humans/rodents) and freely moving rodents, which raises a question of how well the paradigm mimics actual coding of space. This discrepancy should be discussed.

---

## [Author Response]

Task Design:1) The task description requires further detail to understand exactly what was required of participants and, in turn, how to interpret the results. It would be useful to share the specific decisions each participant had to make for each level of spatial scale, or at the very least, examples from each level of spatial scale. An example at the room level is given, but similar in-text samples of at least one block per spatial scale is necessary for the reader to have a better handle on the actual decisions the participants made. Consider adding a short description of the task before the results, and ideally a Methods figure in the main text accompanied by a set of example cues and trials from each of the 6 spatial scales.

As suggested, we have added this to the main text as the new Figure 1, accompanied by example stimuli from one location in each spatial scale. We describe the task and stimuli in this figure’s legend and more extensively in the Materials and methods section, and refer to the figure in the Introduction.

2) While the increasing levels of spatial scale make intuitive sense, the cut-off between each category seems rather arbitrarily defined. For example, it was not clear why the spatial scales progressed from house to neighbourhood when an interim level of street could be made. Similarly, the leap from city to country further seemed to neglect intermediate spatial scales such as region/county etc. It would be helpful for the authors to acknowledge or justify the use of these seemingly arbitrary cut-off points between these spatial scales and to perhaps consider how a more granular classification system might result in different findings.

The main theme of the manuscript is, as stated by the reviewers, the continuous shift in processing along cortical gradients with increasing spatial scale, regardless of granularity. Our original reasoning for the design was that for balancing reasons, we wanted to have two levels in the small scale (room, building), two in the middle (neighborhood, city), and two in the large spatial scale (country, continent). We further attempted to use ecologically valid scales that subjects naturally refer to, and therefore did not include counties (as our subjects live in Israel, which is not divided into counties due to its relatively small size).

To quantitatively investigate the cut-offs between scales and justify them, we have now explicitly attempted to measure the size of each scale across participants. We did this by identifying the latitude and longitude coordinates for each of the provided stimuli, for scales where these referred to identifiable locations (the neighborhood, city, country and continent scales). We managed to identify 72% of the 1824 locations provided by the subjects and measured the distances between all of the locations within each scale. On average, the distances were 350m between locations in neighborhoods, 2.8km between locations in cities, 233km between locations in countries and 1,140km between locations in continents. Assuming that the average distance between elements in a room is ~1m and between elements in a building ~10m, the scales in our design present a relatively stable logarithmic increase (r^2^ of a linear fit to the logarithmic values = 0.98).

We have now added this analysis to the Materials and methods section, and added the figure as Figure 1—figure supplement 1. We further write in subsection “Experimental stimuli”: “These scales reflect ecological categories, which grow in size in a logarithmic manner (Figure 1—figure supplement 1)”. We also write in the manuscript Discussion section: “to obtain a large range of spatial scales and maintain ecological validity we used a personalized paradigm where subjects provided names of real-world locations familiar to them, in six naturalistic scales, therefore not controlling for the precise size and distances in each scale. Despite this restriction, the distances between subjects’ selected stimuli logarithmically increased with each scale, and a bilateral gradient organization was consistently observed across gradients. However, the exact relations between distances and scales may be further investigated in a more granular manner using studies of well-controlled (e.g. virtual) environments with different scales”.

3) The specific composition of object pairs in the distance judgments task is not clearly presented. From my understanding, participants provided 2 locations per level of spatial scale (total = 12 locations), and then produced 8 relevant items each. Then the task itself had 4-5 runs that contained 4 blocks per spatial scale (total = 16 blocks). Each block then contained 4 stimulus pairs and a target/cue, anchoring the participants' decisions. Therefore, there should be at least 4*4*4 (# of runs * # of blocks * # of pairs) pairs of items for each spatial scale across the experiment.

We thank the reviewers for this clarification. There were indeed two locations for each of the six spatial scales (total = 12 locations), and eight items in each location. In each experimental run, there were two blocks for each of the twelve locations (total = 24 blocks per run), and each block included four different stimulus pairs. Therefore, there were 4-5 runs * 24 blocks * 4 pairs = 384-480 pairs of comparisons for each participant. These details are now clarified in the Materials and methods section of the revised manuscript: “subjects were asked to provide names of two real-world locations personally familiar to them at each scale […] In total, subjects performed 24 blocks per run, each including four object pairs, for a total of 384-480 comparisons over the experiment.”

4) Were all pairs unique for each spatial scale and subject, or were there repetitions?

Pairs of items were chosen at random from each location, and repetitions across the experiment were allowed. We now calculated the number of repeated pairs across the experiment: on average, across subjects, only 3.5% of stimuli pairs were repeated across the experiment with the same anchor stimulus. This is now explicitly mentioned in the revised manuscript’s Materials and methods section: “Anchor items and stimuli pairs were chosen independently and randomly from the eight items the subject provided for each location, allowing for repetitions; on average, 3.5% of stimuli pairs were repeated during the experiment (with the same anchor stimulus).”

5) Were all items drawn exclusively from combinations of the 12 items produced by the corresponding subject?

All stimuli were drawn exclusively from combinations of the 8 items produced by the corresponding subject per spatial location (16 items per scale). This is now explicitly mentioned in the revised manuscript’s Materials and methods section: “Anchor items and stimuli pairs were chosen independently and randomly from the eight items the subject provided for each location”.

6) How was the cued item (e.g., "the bed") selected per block?

The cued item was selected randomly for each block, from the eight items provided by the subject for each location. This is now detailed in the revised manuscript’s Materials and methods section.

7) Were all 12 items produced by each subject used as targets? In pairs? Were they all presented an equal number of times?

As items were chosen at random, they did not have to be presented an equal number of times. Following the reviewer’s comment we have calculated and found that items were used as targets on average 9 ± 2.87 times for each subject. We now detail in the Materials and methods section: “on average, 3.5% of stimuli pairs were repeated during the experiment (with the same anchor stimulus), and each item was used 9 ± 2.87 times as a target.”

8) Were these 12 same items used in the 2-5 minutes training task?

The same items were used in the short training phase, as we now detail in the Materials and methods section. On average, there were 4.1 questions used in the training task that were repeated in the fMRI experiment (~1% of the experiment questions), and therefore we do not expect these to interfere with the test phase results.

9) On how many runs of each spatial scale were the participants trained? Did each training run have the same parameters as the experimental run (e.g., number of trials/object pairs)?

Subjects performed the training until they indicated that they felt comfortable doing the task. On average, they performed 53 ± 26.6 comparisons, or 8.8 comparisons per scale. We now detail this in the methods: “A training task using pairs of stimuli derived from the same pool was delivered before the experiment; subjects performed the training until they indicated that they felt comfortable doing the task (average number of training trials per subject = 53 ± 26.6, or 8.8 trials per spatial scale). “(Subsection “Experimental paradigm”).

10) Were the post-task difficulty judgments done for each specific object pairing, or was a rating produced per individual object? E.g. 1-7 difficulty rating for "Table - Window"? 1-7 difficult rating for "Table"?

The post-task ratings of familiarity, emotion and difficulty were done for each of the 12 provided locations, and the perspective taking ratings and strategy descriptions were done for each of the six scales. The behavioral ratings details are now described in the Materials and methods section: “After the experiment, subjects rated their level of familiarity with each of the twelve locations, the emotional significance of the location, and level of difficulty of judgments at each location (from 1 to 7). They were also asked to describe the strategy used for determining responses in each of the six spatial scales (free descriptions) and specifically to what extent did they adopt a ground-level or bird’s-eye point-of-view (1 to 7 rating)”.

Neuroimaging data:11) What precisely was the modelled predictor? The specific onset of each object pair presentation from each block, per spatial scale, drawn across runs (e.g., stick function)? Or was it a boxcar function that modelled activity for the duration of each block? Please clarify.

We modelled the whole duration of each block as a continuous boxcar function. We now clarify in the revised manuscript that “Each modeled predictor included all experimental blocks at one spatial scale, where each block was modeled as a boxcar function encompassing the target stimulus and the four distance comparisons following it. Predictors were convolved with a canonical hemodynamic response function, and the model was fitted to the BOLD time-course at each voxel.” (subsection “Functional MRI analysis”).

12) Was there any additional nuisance regression that was conducted (e.g., white matter, or cerebrospinal fluid)? If not, why?

According to the reviewers’ suggestion, we have now re-done all of the analyses with addition of average white-matter and CSF signals as nuisance regressors. All of the results and statistically significant effects remain unchanged after adding these regressors. We updated all the figures accordingly and detail in the revised subsection “Functional MRI analysis” that: “white matter and CSF masks were manually extracted in BrainVoyager for each subject (intensity>150 for the white-matter mask and intensity<10 with a bounding box around the lateral ventricles for CSF), and the average signals from these masks were added to the GLM to eliminate potential noise sources.”

13) In terms of analysis, were trends other than linear ever tested? Did the linear fit have the most explanatory power relative to other potential trends (e.g., quadratic, cubic…)?

To detect gradual changes in between scales we used here a linear fit, applied on the scale preference graphs along each of the gradients (Figure 2E). Importantly, this fitting was not attempted to model the precise shape of the scale change across the gradients, but rather to quantitatively measure whether scale preference consistently increases or decreases along the posterior-anterior axis of each gradient. To this aim, we estimated the slope of the increase in preferred scale (using an approximate linear fit), and tested whether it is larger than what would be obtained by chance if there was no actual increase in scale preference (using permutations of the scale preference values). Across the four regions (medial parietal, medial temporal, lateral parietal and hippocampus) and two scale preference measures (maximally active scale and peak of Gaussian fit to the beta values), there was a significant slope deviation from random value permutations (p<0.01 for all regions, FDR-corrected), indicating an increase in spatial scale preference along each gradient. In this regard, using a quadratic or cubic fit cannot provide a measure of overall increase in preferred scale along each gradient, as these do not model a gradual directional change. We now clarify this in the subsection “Measuring increase of scale selectivity along gradients and along the hippocampal long axis”: “To measure whether there is a gradual increase in preferred scale along each gradient, we modeled each gradient using a linear function that was fitted to the scale preference values along it, and also fitted this function to 1000 shuffled versions of each scale preference vector for obtaining a null distribution. The slope of the actual fit was tested against the slope of the fits to the random permutations to check if the obtained increase in scale preference along the gradients significantly deviates from chance. Resulting p-values were corrected for multiple comparisons across gradients using the false discovery rate (Benjamini and Hochberg, 1995).”

14) The behavioral data (Figure 1—figure supplement 2 and Supplementary file 3) should be used to segment the fMRI data, as at least in some cases it could provide an additional explanation to some of the gradients. Specifically:a) When grouping data according to the categories in Supplementary file 3, what does the fMRI signal look like? (e.g. imagining a map-like view vs. triangulation).b) How does the fMRI signal segment when weighing according to Familiarity rating, or Perspective-taking rating? (Figure 1—figure supplement 2).

The reviewers mention that perspective-taking, strategy, emotion, difficulty and familiarity are important factors to consider, and may partially explain why making judgment at different scales activates different regions along the gradients we identified. To investigate these issues more in depth, we have now created parametrically modulated regressors according to each of these factors, with the values derived from our subjects’ ratings: degree of difficulty, emotional valence, location familiarity, first-person and third-person perspective imagination, and whether subjects reported or not using specific strategies in the different scales (imagining lines toward the objects, calculating walking/driving/flying times to each item, imagining themselves “looking around” within the scene, imagining a mental map). Note that for the verbal strategy descriptions we do not have quantifiable data for each subject and scale, as subjects were asked an open question on strategy use and we counted whether they mentioned using this strategy or not. We ran a random-effects GLM analysis for each of these factors without taking into account the spatial scale, to investigate their contribution to the observed effects. We now show the results of each of these GLMs in Figure 2—figure supplement 5.

The results of this analysis show that four factors explain significant variance in parts of the gradients: level of familiarity each location, use of a first-person perspective, use of a third-person perspective, and using a strategy of map imagination. This makes sense as these factors are correlated with the change in spatial scale (average correlation across subjects between linear scale increase and these four factors: r = -0.69, -0.81, 0.75, 0.77). For this reason, regions that show a linear increase or decrease in activity with increase in scale will also show the same effect with relative to these factors that are correlated with scale.

The new analyses and now described in the revised manuscript’s Materials and methods section. For clarity, we combined the subsection “subjects’ ratings and reports” and subsection “ruling out effects of possible confounds” into a new subsection “Subjects’ behavioral ratings and their relation to the scale effects”, in which we detail the specific correlation values between scales and behavioral measures. We further explain “To measure the effect of these different factors on the observed activations, we used parametric modulation using subjects’ ratings of emotion, familiarity, difficulty, perspective taking and strategy. The familiarity, perspective taking (first-person and third-person) and reports of use of a map strategy showed significant effects inside the scale-related gradients, in accordance with their high correlation to spatial scale (Figure 2—figure supplement 5). No other factor showed any significantly active regions in this analysis”. We also detail in the discussion section that “These scale-selective gradients were correlated with a shift from detailed to less-detailed knowledge of locations, and from first- to third-person perspective taking with increasing scale”.

We discuss extensively the contribution of these factors in the discussion, as detailed in the answer to the next comment.

Interpretation:15) A fundamental question relates to the nature of smaller versus larger scale environments. For example, a room is far less dynamic than a city and does not necessarily require any movement or navigation within the spatial array to correctly adjudicate between the two distances. How can we determine whether scale preference is the critical factor versus the type of experience that one has at these different spatial scales? Participants were not asked whether the judgments within smaller scale environments invoked retrieval of specific prior experiences (i.e., episodic memory). Medial temporal and medial parietal activation for small-to-large environments may not necessarily reflect the spatial scale but the harnessing of personally relevant episodic experiences during the task. Please discuss.

There are indeed several factors that vary across scales in real life and may probably account, to a different degree, for the scale-related activity differences reported in our study. Further studies of several of these factors are now carried out in our lab in a fully controlled manner, investigating in detail the contribution of each factor. The current study did not aim at fully disentangling these factors, a task that requires several more dedicated studies, but rather to demonstrate the difference in cortical processing associated with spatial judgments at different scales. As suggested by the reviewers, we discuss these factors in the revised manuscript. These factors include:

Dynamics / movement in each scale – as mentioned, the smallest scale (room) might not require movement to judge spatial relations in it, in contrast to larger scales. We did not find any consistent differences in verbal reports of imagined movement between the scales, but this factor might still play a part in the observed differences.

Different degree of use of personally-relevant episodic memories – while these may indeed differ between scales, the specific study design requiring quick judgements in a very short time (2.5s per stimuli pair) did not leave room for such detailed elaborations. In addition, episodic-autobiographic memories engage prominently the DMN, which we found to be the most active for the largest spatial scales that subjects described as involving more third-person, map-like imagery and not first-person imagination. However, we cannot rule out the effect of this element.

Level of familiarity and first vs. third person perspective taking were shown to correlate with scale (and thus with our results; see comment 14). Previous studies directly manipulating environmental knowledge / familiarity (Epstein et al., 2007a, 2007b, Hirshorn et al., 2012b, Wolbers and Buchel 2005) and perspective taking (Rosenbaum et al., 2004, Sherrill et al., 2013) found that the posterior parts of our gradients (PPA, RSC and OPA) are more active for more well-known locations and first-person perspective, as we find here. However, these studies did not describe the opposite pattern in regions anterior to them for less well-known locations and third-person perspective taking, suggesting that these effects cannot fully account for the gradients we observe. One way to reconcile these issues is to suggest that posterior parts of the gradients contain representations that are highly detailed and related to actual perception of the spatial environment, in accordance with the relation of these regions to the visual system and their experience through a first-person perspective. As the scale increases, activity shifts to anterior DMN regions containing less-detailed representations, which are more schematic / abstract and therefore support a third-person, or map-like, imagination of the environment.

We have now added a paragraph to the Discussion section to discuss these issues: “Several factors might explain the shift in cortical activity when subjects make judgments at different scales. One element that may differ between scales is the amount of movement involved in their navigation and initial learning, although we did not find consistent differences between reports of imagined movement at different scales. […] These findings might be explained by the idea that posterior gradient regions contain detailed spatial information, supported by the visual system and acquired using a first-person perspective; as the scale increases, knowledge becomes less detailed and more abstract and schematic, supporting a bird’s-eye / map-like imagination (Arzy and Schacter, 2019).” (Discussion section).

16) The authors note the recruitment of the DMN for larger spatial scales and suggest the potential relationship with representations of abstract domains more generally. It may be interesting to examine this directly using a tool like Neurosynth (http://neurosynth.org). For example, one could examine the% voxel overlap between whole brain maps at each spatial scale and meta-analytic maps of "abstract" and "concrete" obtained via Neurosynth.

We thank the reviewers for this suggestion, which could potentially significantly contribute to the current paper as well as testing further our hypotheses. We therefore enthusiastically aimed to perform the analysis suggested above. However, regardless of the different scales, the Neurosynth maps for “abstract” and “concrete” were almost completely identical (see Author response image 1), making this database problematic to use:

This similarity seems to be related to the Neurosynth algorithm, which takes all of the activations reported in a study and associates them with the study’s keywords. We have looked at full text version of the most highly weighted individual manuscripts included in Neurosynth for the terms “concrete” and “abstract”, it seems that many of them compare concrete and abstract processing, and therefore include the two keywords and their activations are associated with both. For this reason, calculation of the overlap between scales and neurosynth maps does not yield informative or clear results (see Author response table 1 below). We believe that due to this methodological issue the analysis may be misleading, and therefore prefer not to include it in the revised paper.

**Author response table 1. resptable1:** Neurosynth results – overlap of the “concrete” and “abstract” maps with each scale-specific gradient part.

	Room	Building	Neighborhood	City	Country	Continent
**Parahippocampal gradient**						
Concrete	5.9%	20%	9.9%	5.2%	0%	0%
Abstract	10%	16.8%	1.8%	0%	1.8%	0%
**Retrosplenial gradient**						
Concrete	0%	0%	18%	31.6%	27.5%	15.3%
Abstract	0%	0%	2%	4.4%	19%	16.1%
**Occipito-parietal gradient**						
Concrete	4.8%	16.9%	0%	0%	27.9%	23.6%
Abstract	16.2%	13.7%	0%	0%	23.3%	39.7%

17) In Figure 1 (E, bottom right panel), the hippocampal long axis does not show a clear scale-dependent activity gradient, contrary to the claim in the Discussion section, and to previous works (subsection “Three posterior-anterior gradients of spatial scale selectivity”). Activity spans three scales (neighborhood->country) but seems to be involved mainly with 'city' scale. Furthermore, surprisingly it is not selective at all in 'room'/'building' scale. The above have been well established in both VR (humans/rodents) and freely moving rodents, which raises a question of how well the paradigm mimics actual coding of space. This discrepancy should be discussed.

We thank the reviewers for this question. We have performed this analysis anew, calculating the fit in each subject separately and this time using all of the hippocampal voxels, after the addition of CSF and WM regressors to the GLM as suggested in comment 12. We found that there is a significant shift in activity preference from posterior to anterior hippocampus, both in the peak of Gaussian fit to the voxels, and in the maximally active scale, as indicated by a higher than chance slope of a linear fit to the average scale graph (p=0.004, p=0.001, respectively). We updated Figure 2E (previously Figure 1E) and the Materials and methods section accordingly.

Regarding the issue of hippocampal sensitivity to different scales, this analysis reveals that the hippocampus is most active for judgments at the neighborhood, city and country scales. However, this does not mean that it is not involved in the processing of other conditions, such as room and building; our analysis of scale preference indicates the preferred scale for each voxel / region, and not whether other scales also activate the region. With regard to the hippocampus, its activity across all conditions is negative relative to the baseline, so it is difficult to say if it is significantly active for the smaller scales (despite the significant differences between scales). Negative hippocampal BOLD signal is a common finding in the literature, and has been attributed in the past to it being part of the default-mode network, and thus constitutively active during rest (Ekstrom 2010, Stark and Squire 2001, Shipman and Astur 2008). To reiterate, because of the negative BOLD we can only infer from our data that parts of the hippocampus are active for specific scales more than for others, and not whether they are active for each condition specifically. We now discuss this point in the revised manuscript as follows: “Activity in the hippocampus, and in some of the anterior parts of the cortical gradients, was negative relative to baseline, while showing consistent differences in activity between scales. The anterior parts of the three cortical gradients overlap with the DMN, which may be characterized by negative BOLD during tasks (Raichle et al., 2001), and negative BOLD in the hippocampus is also a common finding (Shipman and Astur, 2008). These negative activations were interpreted in the past as potentially reflecting high constitutive activity of these regions during rest more than during active tasks (Ekstrom, 2010; Shipman and Astur, 2008; Stark and Squire, 2001). The fact that these activations are below baseline preclude inference of whether these regions participate in processing of smaller spatial scales or are only active for larger ones.” (Discussion section).